# Dual nature of magnetic dopants and competing trends in topological insulators

Paolo Sessi[1], Rudro R. Biswas[2], Thomas Bathon[1], Oliver Storz[1], Stefan Wilfert[1], Alessandro Barla[3], Konstantin A. Kokh[4,5,6], Oleg E. Tereshchenko[5,6,7], Kai Fauth[1,8], Matthias Bode[1,8] & Alexander V. Balatsky[9,10]

Topological insulators interacting with magnetic impurities have been reported to host several unconventional effects. These phenomena are described within the framework of gapping Dirac quasiparticles due to broken time-reversal symmetry. However, the overwhelming majority of studies demonstrate the presence of a finite density of states near the Dirac point even once topological insulators become magnetic. Here, we map the response of topological states to magnetic impurities at the atomic scale. We demonstrate that magnetic order and gapless states can coexist. We show how this is the result of the delicate balance between two opposite trends, that is, gap opening and emergence of a Dirac node impurity band, both induced by the magnetic dopants. Our results evidence a more intricate and rich scenario with respect to the once generally assumed, showing how different electronic and magnetic states may be generated and controlled in this fascinating class of materials.

[1] Physikalisches Institut, Experimentelle Physik II, Universität Würzburg, Am Hubland, 97074 Würzburg, Germany. [2] Department of Physics and Astronomy, Purdue University, 525 Northwestern Avenue, West Lafayette, Indiana 47907, USA. [3] Istituto di Struttura della Materia, Consiglio Nazionale delle Ricerche, 34149 Trieste, Italy. [4] V.S. Sobolev Institute of Geology and Mineralogy, Siberian Branch, Russian Academy of Sciences, 630090 Novosibirsk, Russia. [5] Physics Department, Novosibirsk State University, 630090 Novosibirsk, Russia. [6] Physics Department, Saint-Petersburg State University, 198504 Saint-Petersburg, Russia. [7] A.V. Rzanov Institute of Semiconductor Physics, Siberian Branch, Russian Academy of Sciences, 630090 Novosibirsk, Russia. [8] Wilhelm Conrad Röntgen-Center for Complex Material Systems, Universität Würzburg, Am Hubland, 97074 Würzburg, Germany. [9] Institute for Materials Science, Los Alamos National Laboratory, Los Alamos, New Mexico 87545, USA. [10] Nordita, Center for Quantum Materials, Department of Theoretical Physics, KTH Royal Institute of Technology, Stockholm University, Roslagstullsbacken 23, 106 91 Stockholm, Sweden. Correspondence and requests for materials should be addressed to P.S. (email: sessi@physik.uni-wuerzburg.de).

Topological insulators (TIs) represent a new state of matter where electronic properties are not related to any symmetry breaking but dictated by the concept of topology. Beyond their fundamental interest, the separation of electronic states into a gapped bulk and a helical spin Dirac cone at the surface offers great potential for applications. Ultimately, the success of TIs crucially depends on the ability of functionalizing them with well-defined perturbations. Within this framework, magnetic dopants occupy a primary role[1–18]. As they break the time-reversal symmetry that protects topological states, they are expected to modify the Dirac spectrum while at the same time changing the spin-texture[5–8]. This interaction is expected to result in several exotic phenomena with interesting implications in spintronics and magneto-electric applications[1–4]. The recent experimental realization of the quantum anomalous Hall effect confirmed the presence of a gapped Dirac state[3,4]. On the other hand, the majority of photoemission[9–13] and local tunnelling data[14–18] clearly demonstrate the presence of a finite density of states (DOS) near the Dirac node in magnetically doped TIs. The disorder induced by the dopants further complicates the story as it generates spatial fluctuations of the chemical potential[16], which significantly broaden the lineshape of photoemission spectra potentially impeding the detection of a gap[9–13]. Recent theoretical predictions suggested that the resolution of this apparent paradox might be directly linked to the dual nature of the electronic states generated by magnetic impurities[19]. However, these claims escaped experimental verification.

In the following, we provide compelling evidence that on one hand the emergence of ferromagnetism gaps the Dirac states and on the other hand impurity-induced resonances form an impurity band at or near the Dirac node filling the gap. Both effects originate from the same magnetic impurity doping, leading to the emergence of a two-fluid behaviour. Our results resolve long-standing contradictions in the field. They paint a more complicated and multifaceted story of magnetically doped TIs and provide a detailed microscopic picture that allows engineering electronic and magnetic properties by acting on the delicate balance in between the impurity band and the gapped Dirac states.

## Results

**Structural and magnetic properties.** Figure 1a shows a photographic image representative of the bulk single crystals used in the present study, that is, vanadium (V)-doped $Sb_2Te_3$. This system recently led to the robust experimental realization of the quantum anomalous Hall effect[4], thus being an optimal platform to investigate the interaction of topological states with time-reversal symmetry breaking perturbation. The nominal stoichiometry corresponds to $Sb_{1.985}V_{0.015}Te_3$. Consequently, within the typical quintuple-layered structure characterizing these materials, V atoms substitute Sb as sketched in Fig. 1b at a concentration of 0.75% in each Sb plane. Figure 1c shows a typical large-scale topographic image of the samples acquired by scanning tunnelling microscopy (STM) in the constant-current mode.

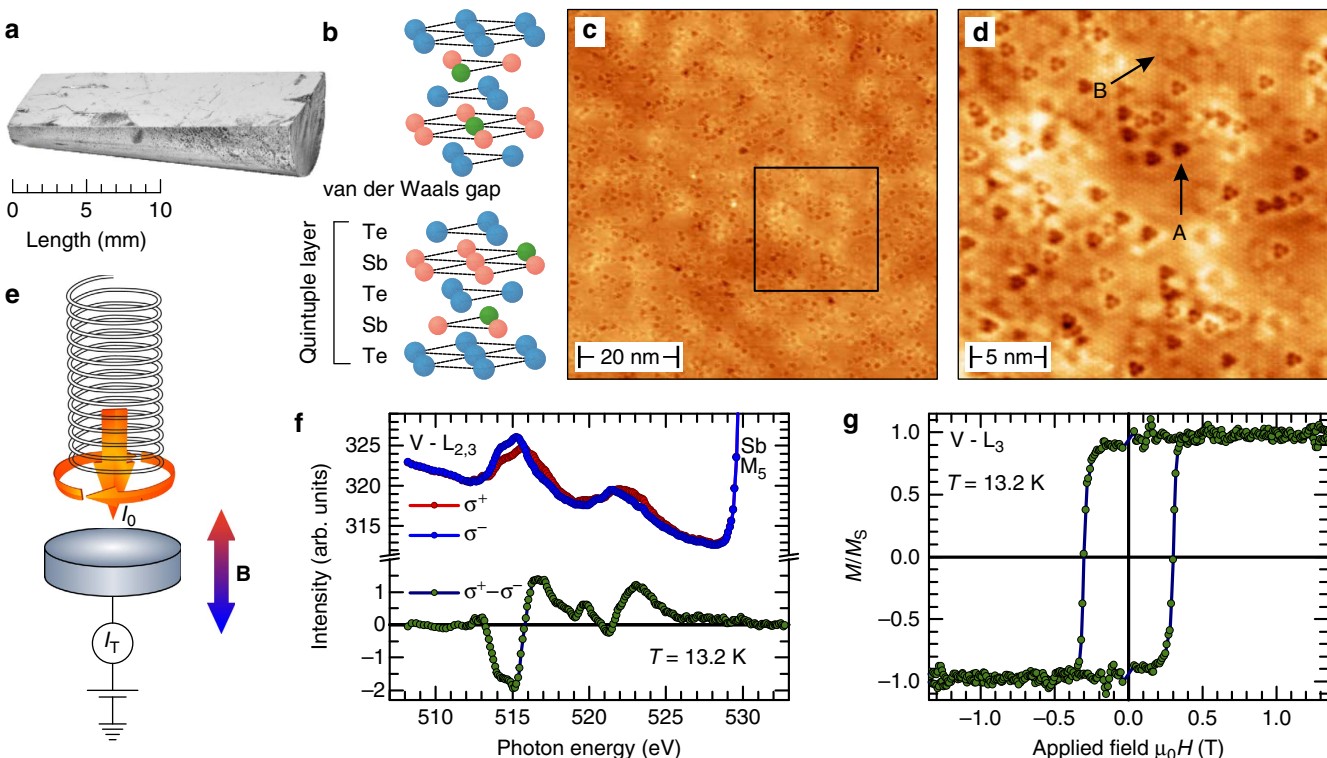

**Figure 1 | Structural and magnetic properties of V-doped $Sb_2Te_3$.** (**a**) Vanadium bulk-doped $Sb_2Te_3$ single crystal ($Sb_{1.985}V_{0.015}Te_3$); (**b**) $Sb_2Te_3$ crystal structure: V atoms (green) are located in the second and fourth layer, substituting Sb; (**c**) $75 \times 75\, nm^2$ topographic image showing the absence of any clustering; (**d**) high-resolution image showing the V substitutional sites. Triangular depressions labelled as A and B correspond to the two possible V substitutional sites located in the second and fourth layer, respectively; (**e**) schematic of XMCD measurements. Photons with well-defined circular polarization and energy are absorbed by the sample in a variable magnetic field. The resulting photocurrent (total electron yield (TEY)) is measured, making the technique sensitive to surface magnetism. (**f**) TEY signals for opposite magnetic fields (blue and red lines) of $B = \pm 1\,T$. Their difference (green line) sheds light on the magnetic response of the V dopants; (**g**) out-of-plane V-$L_3$ XMCD magnetization curve: the square shape of the hysteresis loop proves the existence of long-range surface ferromagnetic order.

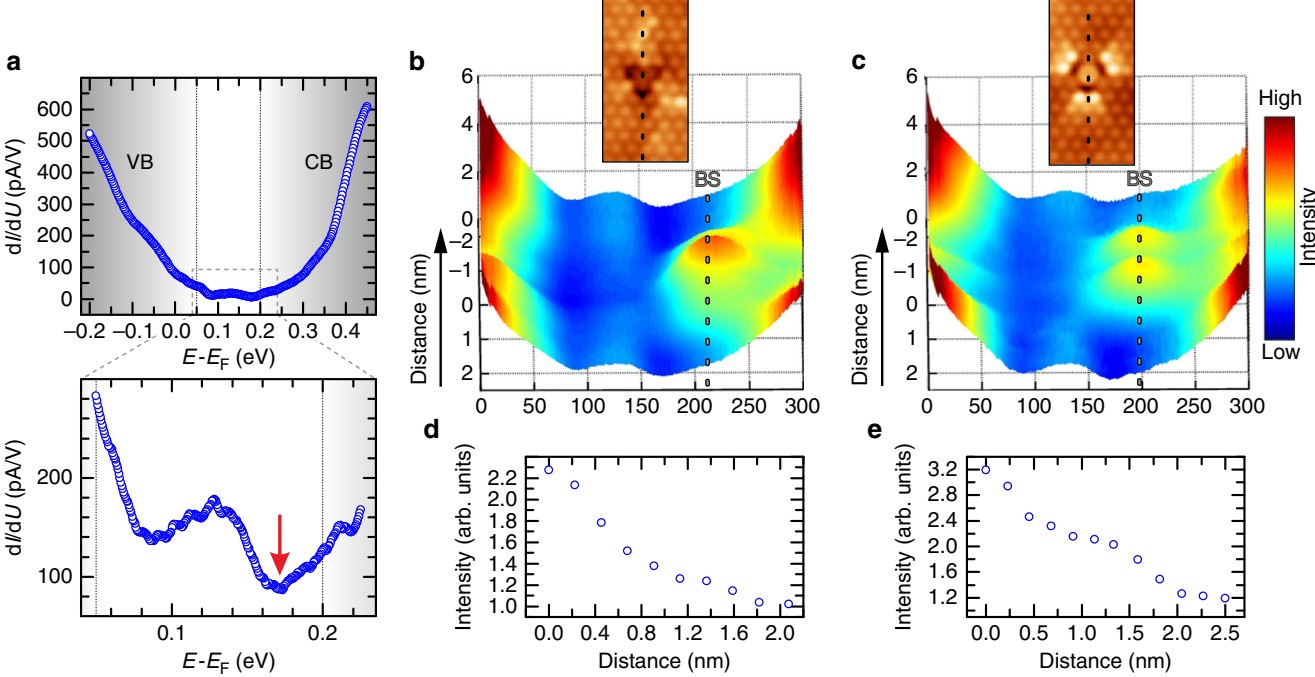

**Figure 2 | Spectroscopic mapping of impurity-induced states.** (**a**) Scanning tunnelling spectroscopy of the ferromagnetic $Sb_{1.985}V_{0.015}Te_3$. Data have been acquired by positioning the tip away from any defect. The density of states reaches a minimum (see arrow) at $\sim 175$ meV above the Fermi level. The spectra reveal the absence of any gap opening despite the existence of long-range ferromagnetic order (see zoomed area). Stabilization parameters: $I = 50$ pA, $V = 450$ mV, zoomed area $I = 50$ pA, $V = 225$ mV; (**b,c**) spectroscopic measurements taken along lines crossing the two different V sites (see insets). In both cases, strong resonances energetically located close to the Dirac point (indicated by BS) emerge in the nearby of the dopants; (**d,e**) spatial dependence of the dopants induced resonances for **b,c**, respectively. The zero in the distance axis corresponds to the position of maximum intensity. In both cases, the spectral weight decays over distances of few nanometres. $T = 4.8$ K.

Dark points correspond to V atoms. Their introduction results in the creation of local perturbations in the crystal structure, which are imaged as depressions at the surface. Note that the V atoms are homogeneously distributed over the surface, without any clustering. This observation proves the good quality of the crystals used in our experiments, which can thus be classified as diluted doped. A quantitative estimation of the V concentration can be achieved by counting the number of depression visible on the surface, giving a value of 0.725% in each Sb layer, in good agreement with the nominal concentration of the crystals. Figure 1d shows an atomically resolved image of the area highlighted by a black box in Fig. 1c. This zoomed image allows visualizing two different V species marked by arrows. An analysis of their appearance based on their symmetry and surface extension allows, as described in ref. 20, to identify them as V atoms located in the second (A) and fourth Sb (B) atomic layer, respectively.

To investigate the magnetic response of the TI to the introduction of magnetic dopants, X-ray magnetic circular dichroism (XMCD) measurements have been performed at the PM3 bending magnet beamline of Helmholtz-Zentrum Berlin HZB[21] with a custom XMCD end station. X-ray absorption was recorded in the total electron yield mode in the presence of an external magnetic field, by measuring the photoinduced sample drain current, as schematically illustrated in Fig. 1e. The short probing depth of this technique[22] allows investigating the existence of magnetic order at the surface, that is, where the topologically protected states exist. Figure 1f reports X-ray absorption spectra taken at the V $L_{2,3}$ edges at a temperature of $T = 13.2$ K in a magnetic field of $\pm 1$ T applied along the surface normal. Their difference ($=$XMCD; green line) highlights the considerable magnetic polarization of the near-surface V dopants

and hence the existence of uncompensated magnetic moments. An XMCD hysteresis curve acquired at the same temperature is reported in Fig. 1g. Its square shape proves the ferromagnetic nature of this polarization and a high degree of homogeneity (absence of domain wall pinning), whereas the full remanence characterizes the surface normal as the easy direction of magnetic anisotropy.

**Impact of impurities on topological states and emergence of Dirac node excitations.** To visualize the impact of impurity ferromagnetism on the topological states, STS measurements have been performed on the very same crystals at $T = 4.8$ K, that is, well below the Curie temperature. The local density of states (LDOS) for energies centred around the Dirac point, that is, where a gap is expected to open, is reported in Fig. 2a by positioning the tip away from any defects. The steep increase of the conductance visible at $\sim 50$ and 200 meV above the Fermi level corresponds to the onset of the valence (VB) and conduction band (CB), respectively. Within this energy range, the conductance is strongly suppressed because of the bulk gap[20]. Spectroscopic measurements optimized over this energy range reveal that, although strongly reduced, the conductance never goes to zero but simply reaches a minimum at $\sim 175$ meV. Unlike in the pristine scenario, where a pure V-shape characterizes STS spectra inside the bulk-gap[20,23], this minimum is energetically close to but not directly at the Dirac point (see below). These data unequivocally demonstrate that, contrary to what is generally assumed, a spectral gap does not open at the Dirac point despite the presence of long-range surface magnetic order, which breaks the time-reversal symmetry protecting topological states.

To shed light on the origin of the finite LDOS, STS measurements have been performed by positioning the tip on top of the V dopants. This allows to directly investigate the V-induced modification of the site-specific electronic properties of the system. In both cases, that is, V atoms occupying the second and fourth layer (Fig. 2b,c, respectively), STS spectra visualize the emergence of strong resonances energetically located close to the Dirac point[24]. Figure 2d,e maps the spatial distribution of their intensity, revealing that their spectral weight is spread over distances of few nanometres, that is, the same length scale as the average distance between dopants (Fig. 1c). These impurity-induced resonances are thus significantly overlapping and likely form an impurity band centred at the resonance energy with a non-negligible spectral weight over the entire surface. Note that the energy position of these resonances is always in close proximity of the Dirac point, that is, where a gap is expected to open because of the magnetically ordered state.

In the following, we demonstrate that their experimental detection is consistent with the expected universal response to perturbations of Dirac materials[25], that is, materials such as d-wave superconductors, graphene and, like in the present case, TIs, whose low-energy excitation spectra is described by massless Dirac particles rather than the Schrödinger equation. This implies that the behaviour of topological states with respect to time-reversal symmetry breaking cannot always be correctly described within the generally assumed framework of gapless versus magnetically gapped states, but that it reflects the competition between the trends for gapping due to magnetic scattering and gap filling due to impurity resonances filling up the gap[6,7,19].

**Dual nature of magnetic dopants: theory discussion.** Once local impurities with both magnetic ($JS$) and scalar potential ($U$) scattering components are introduced in the system[6–7,24,25], the Hamiltonian can be expressed as:

$$\mathcal{H} = \left(k_x \sigma_x + k_y \sigma_y\right) + \sum_{\text{imp}} [J\mathbf{S}_i \cdot \boldsymbol{\sigma}(\mathbf{r}_i) + Un(\mathbf{r}_i)] \quad (1)$$

Here, $J$ and $U$ are the impurity magnetic exchange and potential strengths, respectively, whereas $\mathbf{k}$ is the wave vector. $n(\mathbf{r}_i)$ and $\sigma(\mathbf{r}_i)$ are the electronic number and spin densities, respectively, at the location, $\mathbf{r}_i$, of the $i^{\text{th}}$ impurity. We have chosen units such

that the Planck's constant, $\hbar$, and the Fermi (Dirac) velocity, $v_D$, are both unity.

When the impurities align ferromagnetically perpendicular to the surface (z-direction), it is usually assumed that an energy gap appears at the Dirac point as a result of the spatially averaged carrier–impurity exchange interaction. However, this analysis ignores the role of the impurity potential strength, $U$, which leads to intragap resonances[6,7,19]. Taking this contribution into account, the effective Hamiltonian, including a mean-field gap arising from the magnetic moments, becomes

$$\mathcal{H}_{\text{eff}} = \left(k_x \sigma_x + k_y \sigma_y + \Delta\sigma_z\right) + \sum_{\text{imp}} [J_{\text{eff}}\mathbf{S} \cdot \sigma_z + U]n(\mathbf{r}_i) \quad (2)$$

The mass,

$$\Delta \simeq J \sum_{\text{imp}} \langle n(\mathbf{r}_i)(\mathbf{S}_i)_z \rangle, \quad (3)$$

is half the energy gap of the now massive surface Dirac states, found from an averaging procedure over the impurity-carrier exchange interactions, and $J_{\text{eff}}$ is the residual effective magnetic impurity strength of the impurity acting on these *massive* Dirac fermions. Calculating the exact Green's function of this effective Hamiltonian with the T-matrix method[6] in the presence of a single impurity, we find analytically that when the scalar potential strength, $U$, is larger in magnitude than the residual exchange, $J_{\text{eff}}S$, bound states appear inside the magnetization-induced energy gap. Bound states fill up the spectral gap as shown in Fig. 3. This finding demonstrates that our theoretical approach correctly captures the experimental data (Fig. 2a–e). For a more detailed discussion, see Supplementary Note 1 and Supplementary Figures 1 and 2.

These results paint a more intricate and rich scenario than the general assumption of a spectral gap appearing in magnetic TIs. In particular, they demonstrate the emergence of a two-fluid behaviour induced by the magnetic dopants, which drive a competition in between two different and opposite trends: gap opening versus gap filling which, as in the present case, can coexist.

**Two-fluid behaviour and mobility gap.** This behaviour might sound counterintuitive or even in conflict with the observation of phenomena such as the quantum anomalous Hall effect within this class of systems. In the following, we demonstrate that this is not the case, and that this apparent discrepancy reflects the

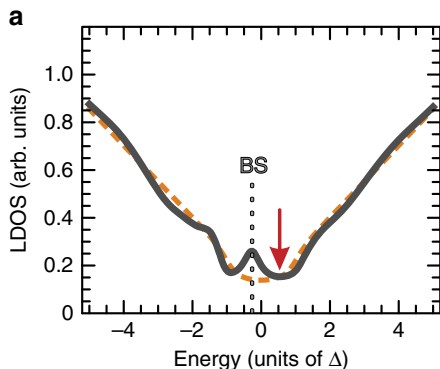
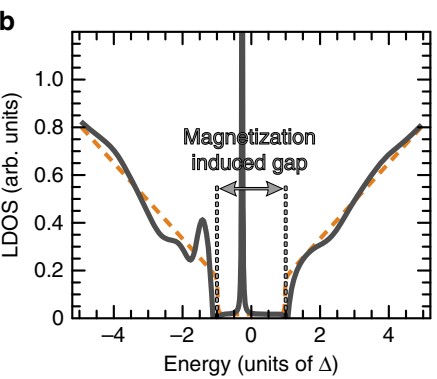

**Figure 3 | Dual-response of topological states to magnetic impurities.** (**a**) Theoretically calculated LDOS at a location with many nearby impurities, using the mean field theory, equation (2). In units where $\hbar$, the Fermi velocity and the gap parameter, $\Delta$, are unity, we have used the impurity strength parameters $J_{\text{eff}}S = 0.5$, $U = 1.75$, high energy cutoff $\Lambda = 50$, and the energy broadening parameter $\Gamma = 0.3$. Impurities are located at distances $r = 0.6, 1.4, 1.5, 1.6, 1.8, 1$ and 3 to mimic their experimental distribution. We have ignored multi-impurity scattering, which would lead to further broadening of the bound state feature (indicated as BS). The dashed curve is the LDOS in the absence of local impurity perturbation. As illustrated in (**b**) for negligible energy broadening ($\Gamma = 0.01$), a clear magnetization-induced energy gap exists between $\pm\Delta$. When the absolute value of $U(J_{\text{eff}}S)^{-1}$ is larger than a threshold (equal to 1 for $\Gamma = 0$), bound states appear inside the gap. Thus, experimental measurements of the LDOS may never show a spectral gap, even though the mobility (transport) gap is nonzero because the localized intra-gap bound states do not affect long range transport at low temperatures.

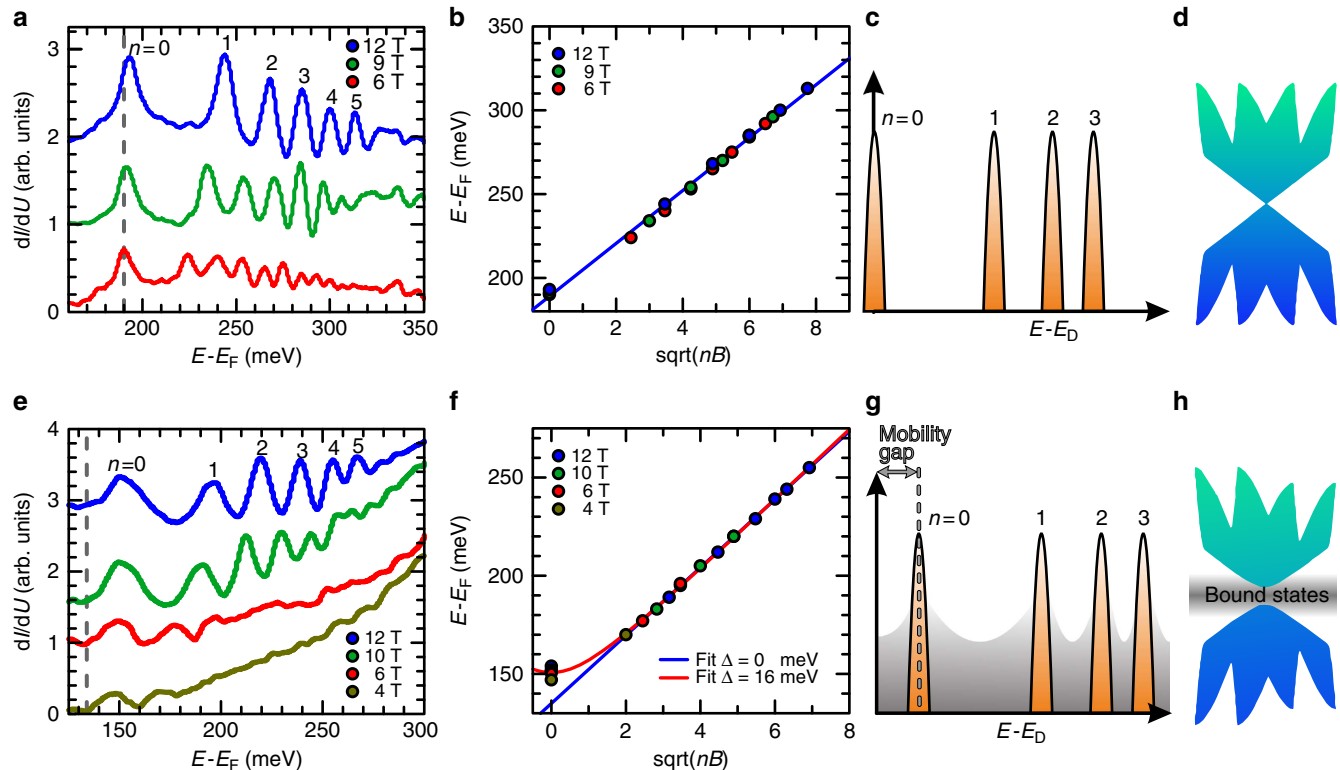

**Figure 4 | Emergence of a two-fluid behaviour.** Landau levels measured by scanning tunnelling spectroscopy in high magnetic fields on a pristine (**a–d**) and (**e–h**) V-doped $Sb_2Te_3$. The spectrum obtained at zero-field has been subtracted in both cases. Spectra at different magnetic fields have been vertically shifted for clarity. The number $n$ identifies the Landau level index. Although all Landau levels of pristine $Sb_2Te_3$ can nicely be fit linearly, this is not the case for $Sb_{1.985}V_{0.015}Te_3$. In this case, the onset of ferromagnetism shifts the zero Landau level towards higher energies, signalling the mass acquisition of the helical Dirac fermions. (**g**) The creation of a mobility gap despite the existence of a finite density of states generated by the magnetic impurities (grey shadow) is illustrated. As schematically illustrated in **h**, these states are filling the gap opened by the broken time-reversal symmetry. Their existence is intimately linked to the dual nature of the dopants, which result in the emergence of a two-fluid behaviour with opposite and competing trends.

localized versus delocalized nature of the impurity induced states, and their different response to electric and magnetic fields. This can be directly visualized by performing Landau level (LL) spectroscopy at high magnetic fields. Figure 4a–d reports the results obtained on pristine $Sb_2Te_3$. For sufficiently strong magnetic fields applied perpendicular to the surface, a sequence of peaks starts to emerge from the background, signalling the condensation of the two-dimensional topological states into a well-defined sequence of LLs[26]. As expected for linearly dispersing states[27], all LLs can be nicely fitted by the relation:

$$E_n - E_D = \mathrm{sgn}(n)\hbar v_D \sqrt{\frac{2e|n|B}{\hbar}} \qquad (4)$$

where $n$ is the LL index, $B$ the magnetic field and $v_D$ the Dirac velocity, which describes the LLs energy sequence for massless particles.

Remarkably, once the same measurements are performed on the magnetically ordered V-doped samples (Fig. 4e–h), the zero-order LL does not follow the fit, but shifts towards higher energies. This is highlighted by the substantial shift of the first peak visible in Fig. 4e with respect to the dashed grey line, which marks the position of the zero-order LL as expected for linearly dispersing Dirac states (raw STS data before zero-field background subtraction are reported in Supplementary Fig. 3). This behaviour unambiguously signals the emergence of a mobility edge for the topological states. The emergence of mobility edges is well-documented in two-dimensional conventional quantum Hall states[28]. In TIs, the evolution of the

LLs for Dirac states, mixed with a continuum of localized states, has the dispersion

$$E_n - E_D = \begin{cases} \mathrm{sgn}(n)\sqrt{2eB\hbar v_D^2|n| + \Delta^2} & , n \neq 0 \\ \Delta & , n = 0 \end{cases} \qquad (5)$$

Here, with respect to the pristine case, the appearance of $\Delta$ signals the acquisition of a mass term for the Dirac states. As shown in Fig. 4f, this equation describes the experimental data very well, the main effect of the mass term being manifested in a shift of the zero-order LL. Concurrently, the magnetic dopants also produce a continuum of localized states as indicated by the shaded regions in Fig. 4g,h. These states also fill the spectral gap about the Dirac point. In agreement with our theoretical arguments, we have found that these are mainly localized states bound to magnetic dopants (Fig. 2d,e). Consequently, once the chemical potential is tuned to their energy range, a mobility gap appears, enabling a full anomalous quantum Hall effect quantization while at the same time inducing a finite DOS at all energies.

## Discussion

In conclusion, we demonstrated that magnetic dopants induce multiple changes with opposite trends in the excitation spectra of TIs. Contrary to what is generally believed, this competition may give rise to the coexistence of long-range magnetic order and gapless states. The microscopic origin of this behaviour is directly linked to the dual nature of the dopants, which on one hand induce a mass term in the Dirac spectrum, whereas at the same time create intragap states with large spectral density within the

magnetic gap. Overall, this delicate balance results in the emergence of a two fluid behaviour where the localization of the impurity resonances produces a mobility gap. Our observations call for a reevaluation of the claims about magnetically doped TI. More generally, they provide evidence of the richness of the physics of TIs demonstrating how, going beyond a single-particle approach, it is possible to tailor the competition between opposite trends paving the way to engineer novel spintronics and magneto-electric effects in this fascinating class of materials.

## Methods

**Crystal growth.** Single crystals of $\sim 10\,mm$ in diameter and $60\,mm$ in length were grown by the modified vertical Bridgman method with rotating heat field[29]. Required stoichiometric amounts of Sb, V and Te were loaded to a carbon-coated quartz ampoule. After evacuation to $p = 10^{-4}\,Torr$ the ampoule translation rate and axial temperature gradient were set to $10\,mm$ per day and $15\,^\circ C\,cm^{-1}$, respectively.

**STM measurements.** Single crystals have been cleaved at room temperature in ultra-high-vacuum and immediately inserted into the STM operated at cryogenic temperatures. All measurements have been performed using electrochemically etched tungsten tips. Spectroscopic data have been obtained using the lock-in technique by adding a bias voltage modulation in between 1 and 10 meV (r.m.s.) at a frequency of $f = 793\,Hz$. Data displayed in Figs 1 and 2 have been obtained using a commercial STM operated at $T = 4.8\,K$. LL spectroscopy data (Fig. 4) have been obtained using a home-built STM operated at $T = 1.3\,K$. The exact energy position of the LLs has been obtained by considering the maximum of a gaussian function fitting each peak after background subtraction.

**Data availability.** The authors declare that the data supporting the findings of this study are available within the article and its Supplementary Information. Data files are available from the corresponding author upon request.

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

## Acknowledgements

Work was supported by the US DOE E304, VR and KAW (A.V.B.) and by DFG through SFB 1170 'ToCoTronics', project A02 (P.S., T.B., O.S., S.W., M.B.). R.R.B. thanks Purdue University Startup funds for support. Allocation of synchrotron radiation beamtime as well as financial support by HZB is gratefully acknowledged. We are grateful to A. Black-Schaffer, J. Fransson and T. Wehling for useful discussions.

## Author contributions

P.S., T.B., O.S. and S.W. performed and analysed STM and STS measurements. T.B., A.B. and K.F. performed XMCD measurements, K.F. analysed them. K.A.K. and O.E.T. synthesized the crystals. R.R.B. and A.V.B. provided the theoretical framework. All authors discussed the results and contributed in writing the manuscript.

## Additional information

**Competing financial interests:** The authors declare no competing financial interests.

