## [Peer Review file · Nature Communications]

Reviewers' Comments:

Reviewer #1 (Remarks to the Author)

The paper by Sessi et al concerns the influence of magnetic defects in topological insulators. They argue that introduction of these impurities has a dual effect on the time-reversal symmetry protected surface states, one is opening of a gap up in the spectrum, due to breaking of time reversal symmetry when defects order magnetically and second is appearance of in-gap states as one would have for any impurity in a semiconductor. These in-gap states make a band according to the authors that fill in the gap, which is the reason why STM spectroscopy or ARPES for that matter has never seen a clean gap for introduction of magnetic impurity into the TIs. A circumstantial case is built around this idea. The experimental data in the paper are proof of magnetism and STM spectroscopy showing no gap and Landau level spectroscopy indicating that the spectrum of Landau levels probing the delocalized states are indeed modified (consistent with a quadratic term and perhaps a gap) when magnetism is introduced in the samples. The authors should perhaps add another effect due to magnetic and non-magnetic impurities that we know also occurs in the TI, which further masks the magnetism induced "gap", this is due to puddling from the long range Coulomb part of the impurity potential. The underlying band structure is fluctuating due to presence of these charged defects in the material. There is STM work that detected this puddling in magnetic and non-magnetically doped TI and showed that they indeed influence the surface states, which the authors should cite (Beidenkopf et al, Nature Physics 2011, I think).

I like this paper and would like to see it published, but I would like to suggest that the authors consider changing the title. I agree with the authors' interpretation of the situation but I would not call this a two-fluid picture. Specially if one of the set of the electronic states is actually localized. The role of defects is indeed complex in a magnetic TI and I think a better title can reflect this new understanding from multiple experiments.

Reviewer #2 (Remarks to the Author)

Dear Editor,

The authors provide a compelling explanation for the fact that often nonzero density of states is measured close to the Dirac point in samples for which one expects (from transport) a Zeeman gap to be present. The idea is to have density of states from immobile carriers due to the same magnetic impurities that create the ferromagnetism.

But in order to be publishable, the evidence for this scenario needs to be made stronger. Figure 2 shows data for the V doped sample, for which the impurity band is resolved. Figure 4 shows that this impurity band leads to a mobility gap for the Landau levels and that this is absent in the pristine case. The readers/reviewers of the paper should be able to verify this point. I.e. the STM spectra have to be shown for the pristine case at zero magnetic field. Is the impurity band indeed absent? Also the data for higher fields should be made available.

Minor comment: the conductance scale in Figure 2a does not match with the zoomed in version. Has a different setpoint been used?

Reviewer #3 (Remarks to the Author)

A

This ms describes measurements taken out on Vanadium bulk-doped Sb₂Te₃ single crystals (Sb_{1.985} V_{0.015} Te₃). Without doping the sample is known to be a topological insulator and the V atoms substitute Sb in the quintuple layer crystal structure that has two Sb layers sandwiched between three Te layers. STM images show a homogeneous atomically flat 75 nm x 75 nm area and close up view reveals two types of defects attributed to V atoms in the second and fourth layer counting from the top layer down towards bulk. As evidenced by means of XMCD, the V atoms exhibit long-range ferromagnetic order with a square hysteresis taken out-of-plane and at 13.2 K ($\mu_0 H_c = 0.3$ T). Scanning tunneling spectroscopy (STS) reveals that there is no gap at E_F and that each type of impurity (V in second and fourth layer) has bound states that decay over a distance of 1 nm. An important part of the paper is the theory. The calculated LDOS shows the magnetization induced gap and bound states for a distribution of magnetic (V) impurities. At finite energy broadening the bound states create DOS in the entire gap opened by the ferromagnetic doping. Therefore DOS measurements don't reveal that there is a gap. However, since the impurities don't participate in transport one has a true transport gap at low T. Landau levels probed by STM substantiate this view. For the pristine Sb₂Te₃, all Landau levels have E proportional to $\sqrt{(n B)}$ while for V doping the zero-order Landau level deviates significantly from the extrapolation of the other levels down to n = 0. This is indicative of the shift of E_{n = 0} with respect to E_D, i.e., it proves the existence of a gap.

B

This ms reports solid results that clarify the issue of band-gap or gap-less magnetically doped topological insulators. It is clearly of general interest and ideal for Nat. Commun. The paper is well written and I recommend it for publication as is, only one minor item has to be checked, see below.

C

approach is very well chosen and presentation is clear

D

yes

E

robust conclusions

F

Check scale for Fig. 2 a. The zoom has the minimum at 80 - 90 pA/V while the large scale spectrum gives the impression that dI/dV goes to 0 at the minimum.

G

yes

H

clear and a pleasure to read

Reviewers' comments:

Reviewer #2 (Remarks to the Author):

Dear Editor,

I am afraid that authors did not correctly understand what data I was asking for. Figure 4 (position of Landau levels) is based on STS spectra taken at fields up to 12 T. In order for me (and the reader) to verify if the determination of the Landau levels from the STS data is unambiguous, one would need to see the actual data (e.g. as supplementary figure).

This is not possible at present, so I cannot recommend publication at this stage.

Reviewers' comments:

Reviewer #1 (Remarks to the Author):

The paper by Sessi et al concerns the influence of magnetic defects in topological insulators. They argue that introduction of these impurities has a dual effect on the time-reversal symmetry protected surface states, one is opening of a gap up in the spectrum, due to breaking of time reversal symmetry when defects order magnetically and second is appearance of in-gap states as one would have for any impurity in a semiconductor. These in-gap states make a band according to the authors that fill in the gap, which is the reason why STM spectroscopy or ARPES for that matter has never seen a clean gap for introduction of magnetic impurity into the TIs. A circumstantial case is built around this idea. The experimental data in the paper are proof of magnetism and STM spectroscopy showing no gap and Landau level spectroscopy indicating that the spectrum of Landau levels probing the delocalized states are indeed modified (consistent with a quadratic term and perhaps a gap) when magnetism is introduced in the samples.

The authors should perhaps add another effect due to magnetic and non-magnetic impurities that we know also occurs in the TI, which further masks the magnetism induced "gap", this is due to puddling from the long range Coulomb part of the impurity potential. The underlying band structure is fluctuating due to presence of these charged defects in the material. There is STM work that detected this puddling in magnetic and non-magnetically doped TI and showed that they indeed influence the surface states, which the authors should cite (Beidenkopf et al, Nature Physics 2011, I think).

Answer: The creation of charge puddles, which might mask the detection of a gap especially by using spatially averaging techniques such as angle-resolved-photoemission is now mentioned as suggested by the referee.

Few additional lines have been included in the introduction:

“The disorder induced by the dopants further complicates the story since it generates spatial fluctuations of the chemical potential [16] which significantly broaden the lineshape of photoemission spectra potentially masking the detection of a gap [9-13].”

The STM work highlighting this effect (Beidenkopf et al, Nature Physics) has now been added as Ref. 16.

I like this paper and would like to see it published, but I would like to suggest that the authors consider changing the title. I agree with the authors' interpretation of the situation but I would not call this a two-fluid picture. Specially if one of the sets of the electronic states is actually localized. The role of defects is indeed complex in a magnetic TI and I think a better title can reflect this new understanding from multiple experiments.

Answer: We decided to change the title that now reads:

“Dual nature of magnetic dopants and competing trends in topological insulators”

Reviewer #2 (Remarks to the Author):

The authors provide a compelling explanation for the fact that often nonzero density of states is measured close to the Dirac point in samples for which one expects (from transport) a Zeeman gap to be present. The idea is to have density of states from immobile carriers due to the same magnetic impurities that create the ferromagnetism.

But in order to be publishable, the evidence for this scenario needs to be made stronger. Figure 2 shows data for the V doped sample, for which the impurity band is resolved. Figure 4 shows that this impurity band leads to a mobility gap for the Landau levels and that this is absent in the pristine case. The readers/reviewers of the paper should be able to verify this point. I.e. the STM spectra have to be shown for the pristine case at zero magnetic field. Is the impurity band indeed absent? Also the data for for higher fields should be made available.

Answer: In a recently published work, we extensively investigated the electronic properties of pristine Sb₂Te₃ (see Phys. Rev. B 93, 035110, now introduced as Ref. 23). Contrary to V-doped Sb₂Te₃ and in agreement with independent experimental findings from other groups (see Ref. 20) no impurity band was found for the pristine case. As suggested by the referee, this information and related references have now been added to the manuscript to let the reader verify this point. Few additional lines have been added to the text which now reads:

“Unlike in the pristine scenario, where a pure V-shape characterizes STS spectra inside the bulk-gap [20,23], this minimum is energetically close to but not directly at the Dirac point (see below).”

Data at higher magnetic fields are not available. The highest magnetic field available in our experimental set-up amounts to 12 T. Data acquired under this condition have been reported in the manuscript. We would like to emphasize that such a magnetic field is among the highest available in scanning probe set-ups (see Rev. Sci. Instrum. 81,121101).

Minor comment: the conductance scale in Figure 2a does not match with the zoomed in version. Has a different setpoint be used?

Answer: The referee is right. To improve the signal-to-noise ratio a different set-point has been used in the zoomed STS spectra. This information has now been added in the caption of Fig.2. We thank the referee for bringing this point to our attention.

Reviewer #3 (Remarks to the Author):

A

This ms describes measurements taken out on Vanadium bulk-doped Sb₂Te₃ single crystals (Sb_{1.985} V_{0.015} Te₃). Without doping the sample is known to be a topological insulator and the V atoms substitute Sb in the quintuple layer crystal structure that has two Sb layers sandwiched between three Te layers. STM images show a homogeneous atomically flat 75 nm x 75 nm area and close up view reveals two types of defects attributed to V atoms in the second and fourth layer counting from the top layer down towards bulk. As evidenced by means of XMCD, the V atoms exhibit long-range ferromagnetic order with a square hysteresis taken out-of-plane and at 13.2 K ($\mu_0 H_c = 0.3$ T). Scanning tunneling spectroscopy (STS) reveals that there is no gap at E_F and that each type of impurity (V in second and fourth layer) has bound states that decay over a distance of 1 nm. An important part of the paper is the theory. The calculated LDOS shows the magnetization induced gap and bound states for a distribution of magnetic (V) impurities. At finite energy broadening the bound states create DOS in the entire gap opened by the ferromagnetic doping. Therefore DOS measurements don't reveal that there is a gap. However, since the impurities don't participate in transport one has a true transport gap at low T. Landau levels probed by STM substantiate this view. For the pristine Sb₂Te₃, all Landau levels have E proportional to $\sqrt{(n+1/2) B}$ while for V doping the zero-order Landau level deviates significantly from the extrapolation of the other levels down to $n = 0$. This is indicative of the shift of $E_n = 0$ with respect to E_D , i.e., it proves the existence of a gap.

B

This ms reports solid results that clarify the issue of band-gap or gap-less magnetically doped topological insulators. It is clearly of general interest and ideal for Nat. Commun. The paper is well written and I recommend it for publication as is, only one minor item has to be checked, see below.

C

approach is very well chosen and presentation is clear

D

yes

E

robust conclusions

F

Check scale for Fig. 2 a. The zoom has the minimum at 80 - 90 pA/V while the large scale spectrum gives the impression that dI/dV goes to 0 at the minimum.

Answer: To improve the signal-to-noise ratio a different set-point has been used in the zoomed STS spectra. This information has now been added in the caption of Fig.2. We thank the referee for bringing this point to our attention.

G
yes

H
clear and a pleasure to read

Reviewer #2 (Remarks to the Author):

Dear Editor,

I am afraid that authors did not correctly understand what data I was asking for. Figure 4 (position of Landau levels) is based on STS spectra taken at fields up to 12 T. In order for me (and the reader) to verify if the determination of the Landau levels from the STS data is unambiguous, one would need to see the actual data (e.g. as supplementary figure).

This is not possible at present, so I cannot recommend publication at this stage.

Answer: We apologize for the misunderstanding. Two additional panels have now been added in Fig.4. These panels report the STS spectra obtained on both pristine and V-doped Sb_2Te_3 . As can be clearly seen, in the V-doped case the energy position of the zero-order Landau level ($n=0$) is significantly higher than the position expected for linearly dispersing Dirac states (identified by a dashed gray line). This observation unambiguously signals the emergence of a mass term.

The exact energy position of the Landau levels has been obtained by considering the maximum of a gaussian function fitting each peak after background subtraction as described in the Methods section.

To further strengthen the robustness of our analysis, and to let the reader verify this point, the raw STS data have been also reported and added into the supplementary material as Fig. S3. Already in the raw data, the zero Landau level in V-doped Sb_2Te_3 significantly shifts with respect to the position expected for linearly dispersing Dirac states.